# *Pdgfra* marks a cellular lineage with distinct contributions to myofibroblasts in lung maturation and injury response

Rongbo Li[1,2], Ksenija Bernau[3], Nathan Sandbo[3], Jing Gu[4], Sebastian Preissl[4], Xin Sun[1,2]*

[1]Department of Pediatrics, University of California, San Diego, La Jolla, United States; [2]Laboratory of Genetics, Department of Medical Genetics, University of Wisconsin-Madison, Madison, United States; [3]Department of Medicine, University of Wisconsin-Madison, Madison, United States; [4]Center for Epigenomics, Department of Cellular and Molecular Medicine, University of California, San Diego, La Jolla, United States

**Abstract** *Pdgfra*-expressing (*Pdgfra+*) cells have been implicated as progenitors in many mesenchymal tissues. To determine lineage potential, we generated *Pdgfra$^{rtTA}$* knockin mice using CRISPR/Cas9. During lung maturation, counter to a prior study reporting that *Pdgfra+* cells give rise equally to myofibroblasts and lipofibroblasts, lineage tracing using *Pdgfra$^{rtTA}$;tetO-cre* mice indicated that ~95% of the lineaged cells are myofibroblasts. Genetic ablation of *Pdgfra$^+$* cells using *Pdgfra$^{rtTA}$*-driven diphtheria toxin (DTA) led to alveolar simplification, demonstrating that these cells are essential for building the gas exchange surface area. In the adult bleomycin model of lung fibrosis, lineaged cells increased to contribute to pathological myofibroblasts. In contrast, in a neonatal hyperoxia model of bronchopulmonary dysplasia (BPD), lineaged cells decreased and do not substantially contribute to pathological myofibroblasts. Our findings revealed complexity in the behavior of the *Pdgfra*-lineaged cells as exemplified by their distinct contributions to myofibroblasts in normal maturation, BPD and adult fibrosis.
DOI: https://doi.org/10.7554/eLife.36865.001

*For correspondence:
xinsun@ucsd.edu

Competing interests: The authors declare that no competing interests exist.

## Introduction

With the increasing understanding of cellular diversity and plasticity at the single cell-resolution, knowledge of the mesenchyme still lags behind the epithelium. *Pdgfra* is expressed in selected cell types in a wide array of mesenchymal tissues, including the lung, heart, intestine, skin and cranial facial mesenchyme (*Boström et al., 1996*; *Chong et al., 2013*; *Karlsson et al., 1999*; *Karlsson et al., 2000*; *Lindahl et al., 1997*; *McCarthy et al., 2016*). In accordance, *Pdgfra*-null mice die at mid-gestation with failed development of multiple mesenchymal tissues (*Soriano, 1997*). Conditional inactivation of *Pdgfra* in the lung mesenchyme at a later time during embryogenesis led to simplified alveoli, suggesting that it is essential for septal ridge formation (*Branchfield et al., 2016*; *McGowan and McCoy, 2014*). In the adult lungs, *Pdgfra$^+$* cells have been described to be in close proximity to alveolar epithelial type 2 cells (AEC2s), and are implicated as niche for AEC2s by way of supporting their proliferation and differentiation (*Barkauskas et al., 2013*). Recent studies further defined the *Pdgfra+* cells as niche for the subset of AEC2s that are especially primed as progenitors in repair (*Nabhan et al., 2018*; *Zepp et al., 2017*). Aberrant PDGFRA activity have been linked to a large number of diseases, including adult diseases such as fibrosis and cancer, and pediatric diseases such as bronchopulmonary dysplasia (BPD) (*Decker et al., 2017*; *Mueller et al., 2016*; *Nannini et al., 2013*; *Olson and Soriano, 2009*; *Popova et al., 2014*). These data support that

*Pdgfra*[+] cells and PDGFRA function in these cells are of key importance in the development, homeostasis and pathogenesis of the mesenchyme.

In the lungs, the term 'myofibroblasts' has been used to define alveolar cells that express smooth muscle markers such as alpha-smooth muscle actin (a-SMA, encoded by *Actin alpha two* or *Acta2* gene), or smooth muscle 22a (SM22a, encoded by *Transgelin* or *Tagln* gene). While lung myofibroblasts most commonly referred to cells in patients with adult diseases such as fibrosis, smooth muscle marker-positive alveolar cells are also found in patients with prematurity related neonatal diseases such as bronchopulmonary dysplasia. Moreover, they are found in normal lungs during maturation. In normal mouse lungs, myofibroblasts are mostly found in the neonatal period, between postnatal day (P) three and P13, in the first phase of alveologenesis (*Branchfield al., 2016*; *McGowan et al., 2008*). After P14, a-SMA and SM22a expression in the alveolar region is drastically reduced to near baseline levels, continuing into the adult. Thus, in the normal lung, the term 'myofibroblasts' refers to the population of cells that are transiently expressing a-SMA+ and SM22a+ durring the process of alveologenesis when secondary septae or crests form. For this reason, they have also been referred to as secondary crest myofibroblasts (*Li et al., 2015*). However, in pathological settings, such as in BPD and neonatal hyperoxia mouse model of BPD, smooth muscle markers continue to be detected in the alveolar region, indicating persistence of myofibroblasts. In the adult lung fibrosis and bleomycin mouse model of fibrosis, smooth muscle markers return in the alveolar region, either from re-expression of these markers in cells that used to express them, or from de novo expression in cells that were never myofibroblasts. The cellular relationship as well as molecular similarities among these three types of myofibroblasts have not been defined. Understanding the mechanisms underlying the dynamic nature of normal myofibroblasts may inform how we can convert fibrotic myofibroblasts into non-fibrotic lung mesenchymal cells.

In the normal lungs during alveologenesis, using a BAC transgenic *Pdgfra-creERT2* line in lineage tracing, it was reported that *Pdgfra*+ cells give rise to both myofibroblasts and lipofibroblasts, with a slight bias towards more lipofibroblasts (*Ntokou et al., 2015*). At this early postnatal stage, myofibroblasts are found to underline nascent septal ridges, suggesting that they may be important for septae formation (*Branchfield et al., 2016*). They also deposit extracellular matrix, such as elastin, which provides the lungs with its elasticity and tensile strength (*Toshima et al., 2004*). In comparison, lipofibroblasts are characterized by their lipid-filled vesicles, and expression of markers such as Perilipin 2 (PLIN2, also known as ADRP). In addition to *Pdgfra*+ cells, it has also been shown that *Tbx4* and *FGF10* expressing cells of the distal lung can give rise to lipofibroblasts during development (*El Agha et al., 2014*; *Zhang et al., 2013*). In the adult lungs, using the BAC transgenic *Pdgfra-creERT2* line, it was shown that lineaged cells do contribute to fibrotic myofibroblasts (*Zepp et al., 2017*). In addition, other cell lineages, for example, *Adrp*-expressing lipofibroblasts can also contribute to fibrotic myofibroblasts (*El Agha et al., 2017*).

In this study, we addressed the relationship between *Pdgfra*+ cells and the three types of myofibroblasts by generating a novel *Pdgfra*[rtTA] knockin mouse line using CRISPR/Cas9 technology. This line faithfully labeled *Pdgfra*-expressing cells. In normal development, we found that *Pdgfra*+ cells give rise primarily to alveolar myofibroblasts and very few lipofibroblasts, counter to previous report (*Ntokou et al., 2015*). In the two models of lung diseases, *Pdgfra*+ cells showed substantial contribution to myofibroblasts in the bleomycin model of fibrosis, but not in the hyperoxia model of BPD. These findings demonstrate that the *Pdgfra*+ cells have distinct lineage potential in different settings, and make distinct contribution to the three types of myofibroblasts.

## Results

### Efficient generation of the *Pdgfra*[rtTA] knockin line using CRISPR/Cas9

To generate *Pdgfra*[rtTA] line, we used the CRISPR/Cas9 technology and designed a guide RNA (gRNA) targeting a site in the first intron of *Pdgfra*. This is the same insertion site as in the *Pdgfra*[GFP] line, which is frequently used as a reporter of *Pdgfra* expression (*Chen et al., 2012*; *Green et al., 2016*; *Hamilton et al., 2003*; *McGowan and McCoy, 2014*; *McGowan and McCoy, 2015*). We chose to generate an rtTA knockin line rather than a creERT2 knockin line for two reasons. First, rtTA is activated by doxycycline which does not induce the frequent abortion observed with prenatal administration of tamoxifen that is needed to induce creERT2. Second, rtTA can be used to turn on/

off transgene expression dynamically, while creERT2 acts through DNA re-arrangements that are not reversible. A double-stranded donor plasmid was generated that contains a splicing acceptor (SA), rtTA and triple polyadenylation sequence (3 PA), with ~1.5 kb homologous arms on each side (*Figure 1A*). Coinjection of CAS9 protein, gRNA and donor plasmid yielded 23 pups. Among them, two had the identical correct rtTA insertion through homologous recombination. Initial characterization indicated that the two lines behaved similarly. Thus, we used one of the two lines for all following experiments (*Figure 1A* and *Figure 1—figure supplement 1A,B*).

## PDGFRa*rtTA* drives rtTA activity faithfully in the *Pdgfra* expression pattern

To determine rtTA activity, we mated *Pdgfra*<sup>rtTA</sup> mice with *tetO-GFP* reporter mice. Doxycycline (dox) administration started at embryonic day (E) 9.5 and continued on to postnatal day (P) 7, the peak of alveologenesis, when the activity was analyzed. In sections of the alveolar region, GFP signal was found regularly at septal tips (*Figure 1B*). This pattern replicates that of the *Pdgfra*<sup>GFP</sup> line. Quantification across sections from multiple lung samples showed that GFP+ cells made up a similar percentage of total alveolar cell population in *Pdgfra*<sup>rtTA</sup>;*tetO-GFP* lungs as compared to *Pdgfra*<sup>GFP</sup> lungs (*Figure 1C*). Quantification using flow cytometry analysis from dissociated lungs revealed that GFP+ cells made up a smaller percentage of total lung cells in *Pdgfra*<sup>rtTA</sup>;*tetO-GFP* lungs as compared to *Pdgfra*<sup>GFP</sup> lungs, but the difference is not statistically significant (*Figure 1D,E*). For labeling efficiency in *Pdgfra*<sup>rtTA</sup>;*tetO-GFP*, flow cytometry analysis showed that out of the total PDGFRa+ cells as labeled by anti-CD140a (PDGFRa)-PE antibody, 78.62 ± 5.71% were GFP+ (*Figure 1—figure supplement 1C,D*). These results indicate that the knockin rtTA strain recapitulates *Pdgfra* expression efficiently.

## Lineage tracing revealed that *Pdgfra* cells give rise primarily to normal alveolar myofibroblasts

To perform lineage tracing of *Pdgfra*-expressing cells, we generated *Pdgfra*<sup>rtTA</sup>;*tetO-cre;Rosa-tdTomato* reporter mice. First, with no dox treatment, only rare tdTomato+ cells were observed, suggesting that there is very little leaky activity (*Figure 2—figure supplement 1A*). Second, when induced with dox starting at E9.5 and analyzed at E12.5, *Pdgfra*+ lineaged cells represent a large proportion of mesenchymal cells, similar to that of GFP+ cells in E12.5 *Pdgfra*<sup>GFP</sup> lungs (*Figure 2—figure supplement 1B,B', C and C'*). However, there are unlabeled mesenchymal cells in lungs of both strains.

Third, when induced with dox starting at E9.5 and continuing on to P7, the peak of alveologenesis, tdTomato +*Pdgfra* lineaged cells were primarily located at the septal tips (*Figure 2A*). TdTomato signal showed substantial colocalization with myofibroblast marker SM22a staining in the alveolar region (*Figure 2B–E*, arrows). A similar colocalization was observed with another myofibroblasts marker, a-SMA (*Figure 2—figure supplement 1D–F*). Quantification revealed that 94.3 ± 1.83% of tdTomato+ cells were SM22a-positive (*Figure 2N*). In comparison, only 5.6 ± 1.19% of tdTomato + cells were positive for the lipofibroblast marker ADRP (*Figure 2F–I* and *Figure 2N*). Interestingly, 3.78 ± 1.48% of tdTomato+ cells were positive for the endothelial marker EGR (*Figure 2J–M* and *Figure 2N*). This was also confirmed with ICAM2 staining, a second marker of endothelial cells (*Figure 2—figure supplement 1G–I*). Focusing on myofibroblasts, quantification revealed that 92.71 ± 0.84% of SM22a + cells were tdTomato+ (*Figure 2O*), indicating that the majority of the myofibroblasts at P7 are derived from *Pdgfra*+ cells. There was no labeling of the vascular smooth muscles at P7 (*Figure 2—figure supplement 1J–L*). While there were labeled cells subjacent to the airway, they did not overlap with airway smooth muscles upon close inspection (*Figure 2—figure supplement 1J–L*).

Fourth, we traced the postnatal lineage starting at birth by a single dose of dox injection at P0 followed by dox food until analysis at P7 (*Figure 2—figure supplement 2A*). Similar to our results from prenatal tracing, 95.57 ± 1.65% of tdTomato+ cells were positive for SM22a, and 5.25 ± 0.75% were positive for ADRP (*Figure 2—figure supplement 2B–I* and *Figure 2—figure supplement 2N*). Distinct to the result from prenatal tracing, no tdTomato+ cells were positive for ERG (*Figure 2—figure supplement 2J–M* and *Figure 2—figure supplement 2N*). Taken together, the findings from our knockin line demonstrate that *Pdgfra*-lineaged cells give rise primarily to myofibroblasts, with minor contributions to lipofibroblasts and endothelial cells.

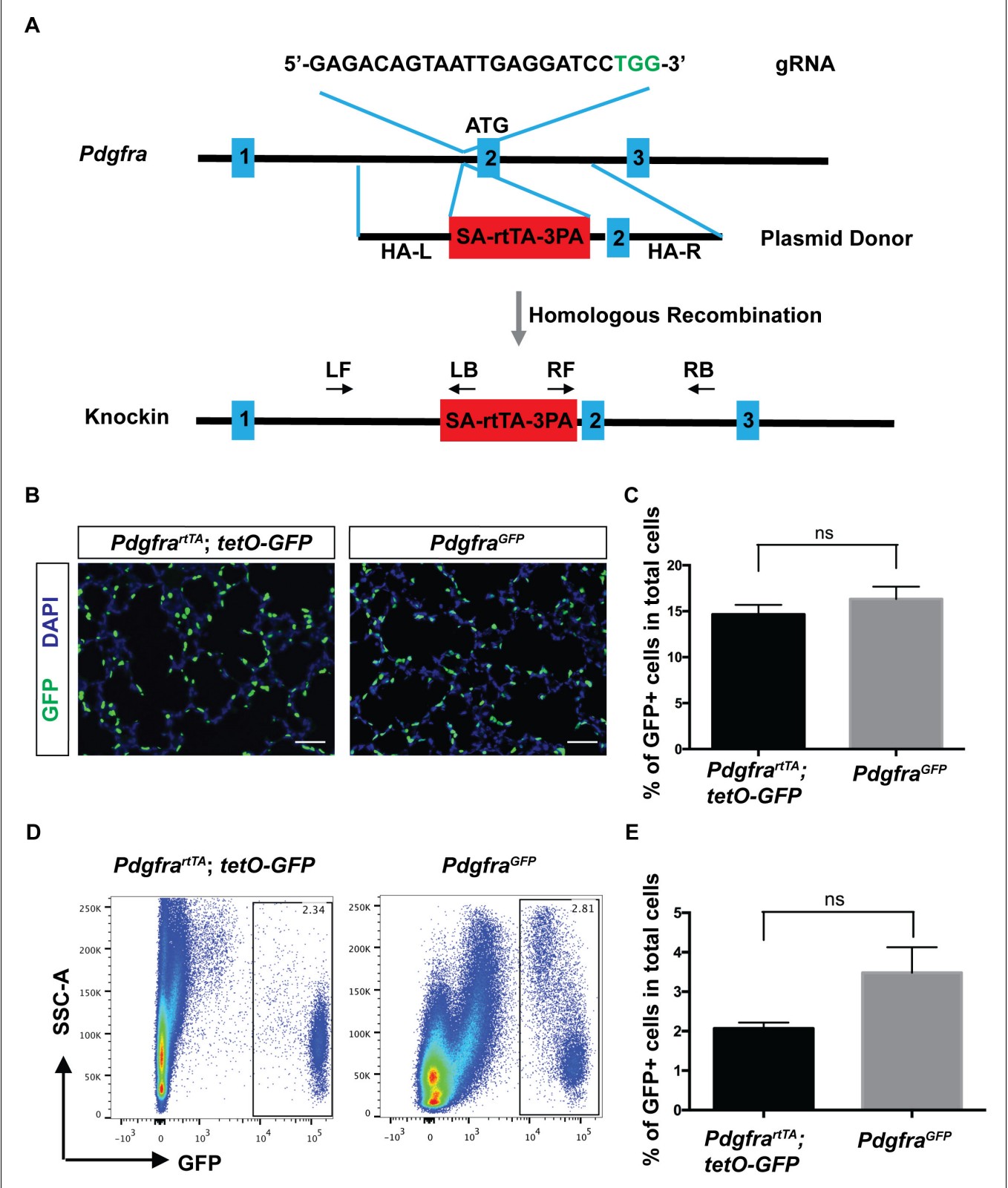

**Figure 1.** Generation and characterization of *Pdgfra^rtTA* knock-in mouse line using CRISPR/Cas9. (**A**) A schematic overview of the strategy for generating the *Pdgfra^rtTA* knock-in allele. The gRNA coding sequence is as shown, and the protospacer-adjacent motif (PAM) sequence is indicated in green. The first three of *Pdgfra* exons are diagramed in blue. The SA-rtTA-3PA fragment is diagramed in red. The homologous arms of the donor vector are indicated as HA-L (~1.5 kb) and HA-R (~1.5 kb). The approximate positions of primers used for PCR analysis are shown (LF-left forward, LB-

*Figure 1 continued*

left back, RF-right forward, RB-right back). (**B**) Representative immunofluorescent staining showing GFP+ cells in *Pdgfra^rtTA;tetO-GFP* mice and *Pdgfra^GFP* mice at P7, scale bars: 50 µm. (**C**) Quantification of the percentage of GFP+ cells as compared to total cell number in the lungs of the two different strains as shown in B (14.66 ± 1.04 for *Pdgfra^rtTA;tetO-GFP* mice and 16.33 ± 1.34 for *Pdgfra^GFP* mice, ns for not significant, p=0.379, n = 3 each). (**D**) FACS analysis of GFP+ cells in *Pdgfra^rtTA;tetO-GFP* mice and *Pdgfra^GFP* mice as indicated at P7. (**E**) Quantification of the percentage of GFP+ cells as compared to total cell number in the lungs of the two different strains as shown in D (2.07 ± 0.15 for *Pdgfra^rtTA;tetO-GFP* mice and 3.48 ± 0.64 for *Pdgfra^GFP* mice, ns for not significant, p=0.0759, n = 4 each).

DOI: https://doi.org/10.7554/eLife.36865.002

The following source data and figure supplements are available for figure 1:

**Source data 1.** Raw data for *Figure 1*.
DOI: https://doi.org/10.7554/eLife.36865.005

**Figure supplement 1.** PCR confirmation of the correct homologous recombination following CRISPR/Cas9 injection and the labeling efficiency of *Pdgfra^rtTA* mice.
DOI: https://doi.org/10.7554/eLife.36865.003

**Figure supplement 1—source data 1.** Raw data for *Figure 1—figure supplement 1*.
DOI: https://doi.org/10.7554/eLife.36865.004

We recently showed that in a normal lung, myofibroblasts as defined by a-SMA and SM22a expression, are transiently detected starting at P3, peaks at P7, lowers to close to undetectable levels starting at P14, continuing into adult (*Branchfield et al., 2016*). The duration of expression corresponds precisely with the first of the two phases of alveologenesis (*Schittny et al., 2008*). This observation raised the question of whether the loss of myofibroblast markers is due to myofibroblast cell death or their downregulation of smooth muscle characteristics. Since *Pdgfra+* cells give rise primarily to myofibroblasts as demonstrated above, we asked whether the tdTomato+ cells simply disappeared, or survived but turned off SM22a expression. After dox induction in the early postnatal stage, tdTomato+ cells were found to survive to adult stage as analyzed at P40, even though very little SM22a or ADRP expression can be detected (*Figure 2—figure supplement 3*). These data suggest that the transient nature of myofibroblasts during alveologenesis is due to these cells turning on and off smooth muscle markers.

## Single cell transcriptome analysis of *Pdgfra+* cells revealed diversity during alveologenesis

To further characterize *Pdgfra+* cells during alveologensis, we performed single-cell RNA sequencing (scRNA-Seq) using fluorescence-activated cell sorting (FACS) enriched GFP+ cells from *Pdgfra^GFP* lungs at P7 (*Figure 1D*) and P15 (*Figure 3—figure supplement 1*). For P7, we analyzed profiles of 3204 sorted cells, with a median of 1596 genes per cell. t-distributed stochastic neighbor embedding (tSNE)-based plot revealed that *Pdgfra+* cells from P7 lungs can be separated into five clusters (*Figure 3A*). Marker gene expression showed that the large cluster one contained the majority of *Pdgfra*-high expression cells, while the large cluster two contained the majority of *Pdgfra*-low expression cells. Interestingly, cluster two exhibited enriched expression of *Fgf10* and *Wnt2* (*Figure 3B,C*). Consistent with prior publication that *Fgf10* lineaged cells give rise to lipofibroblasts (*El Agha et al., 2014*), cluster two contains cells that express *Plin2*, a lipofibroblast marker. Cluster three exhibited enriched expression of genes associated with cell cycle and cell proliferation, such as *Cks2* and *Top2a* (*Figure 3B*). Cluster 4, a distinct small population of cells, expressed *Lgr6*, which is primarily associated with airway smooth muscle cells (*Lee et al., 2017*) (*Figure 3B,C*). Consistent with our finding that *Pdgfra+* cells do not contribute to airway smooth muscles, close examination indicated that cluster 4 cells showed no or low *Pdgfra* expression, suggesting that this small population may have come through GFP+ sorting as escapers from FACS gating. Finally, cluster 5, a distinct small population, exhibited expression enriched in matrix associated genes, including *Decorin* (*Dcn*) and several collagen genes (*Col1a1, Col3a1* and *Col14a1*) (*Figure 3B,C*). Focusing on myofibroblasts and lipofibroblasts, data from scRNAseq suggest that the majority of *Pdgfra+* cells express *Acta2* and *Tagln*, while a minority of them express *Plin2* (*Figure 3C*), consistent with our lineage tracing data.

For P15 *Pdgfra^GFP* lungs, we were able to profile 1,234 cells with a median of 1441 genes per cell. tSNE-based plot revealed that these cells can be separated into four main clusters with a few

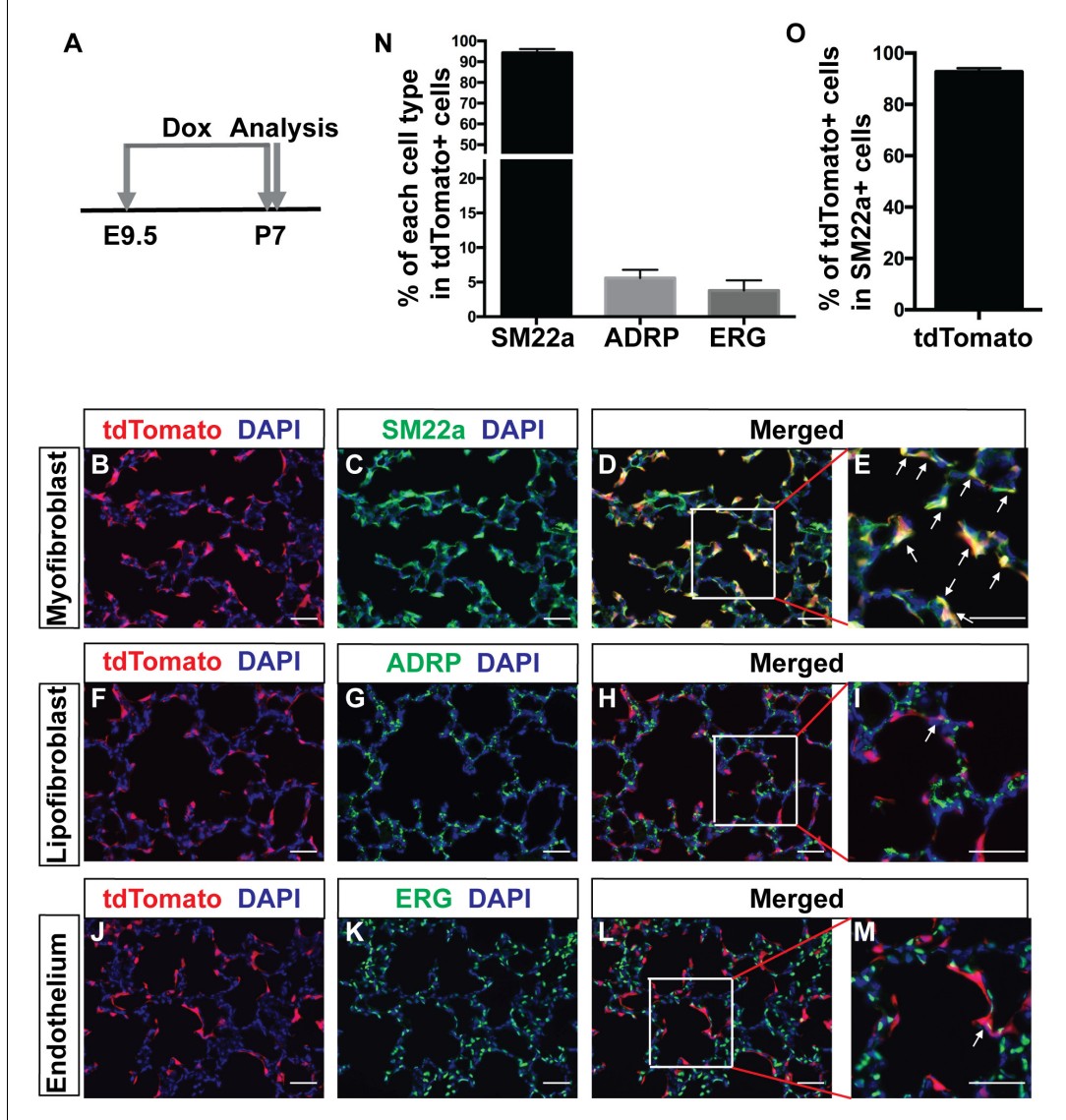

**Figure 2.** Lineage tracing of *Pdgfra* cells starting prenatally. (**A**) Timeline of the start of dox treatment and analysis. (**B–M**) Representative immunofluorescent staining of P7 *Pdgfra^rtTA;tetO-cre;Rosa-tdTomato* lungs. Markers are as indicated: tdTomato (red), the myofibroblast marker SM22a (green) and DAPI (blue) (**B–E**); the lipofibroblast marker ADRP (green) (**F–I**); the endothelium cells marker ERG (green) (**J–M**). Arrows in panels E, I and M indicate cells with co-localization of signals. Scale bars: 50 µm. (**N**) Quantification of the percentages within tdTomato+ cells that are also SM22a+ for myofibroblasts, ADRP+ for lipofibroblasts, and ERG+ for endothelial cells, respectively (94.3 ± 1.83 for SM22a, 5.6 ± 1.19 for ADRP and 3.78 ± 1.48 for ERG, n = 3 each). (**O**) Quantification of the percentage within SM22a+ cells that are also tdTomato+ for lineaged cells (92.71 ± 0.84, n = 3).
DOI: https://doi.org/10.7554/eLife.36865.006

The following source data and figure supplements are available for figure 2:

**Source data 1.** Raw data for *Figure 2*.
DOI: https://doi.org/10.7554/eLife.36865.011
**Figure supplement 1.** Tracing of *Pdgfra*-lineaged cells in prenatal stage.
DOI: https://doi.org/10.7554/eLife.36865.007
**Figure supplement 2.** Tracing of *Pdgfra*-lineaged cells in the early postnatal stage.
DOI: https://doi.org/10.7554/eLife.36865.008
**Figure supplement 2—source data 1.** Raw data for *Figure 2—figure supplement 2*
DOI: https://doi.org/10.7554/eLife.36865.009
**Figure supplement 3.** Lineage tracing of PDGFRa cells to adult.
DOI: https://doi.org/10.7554/eLife.36865.010

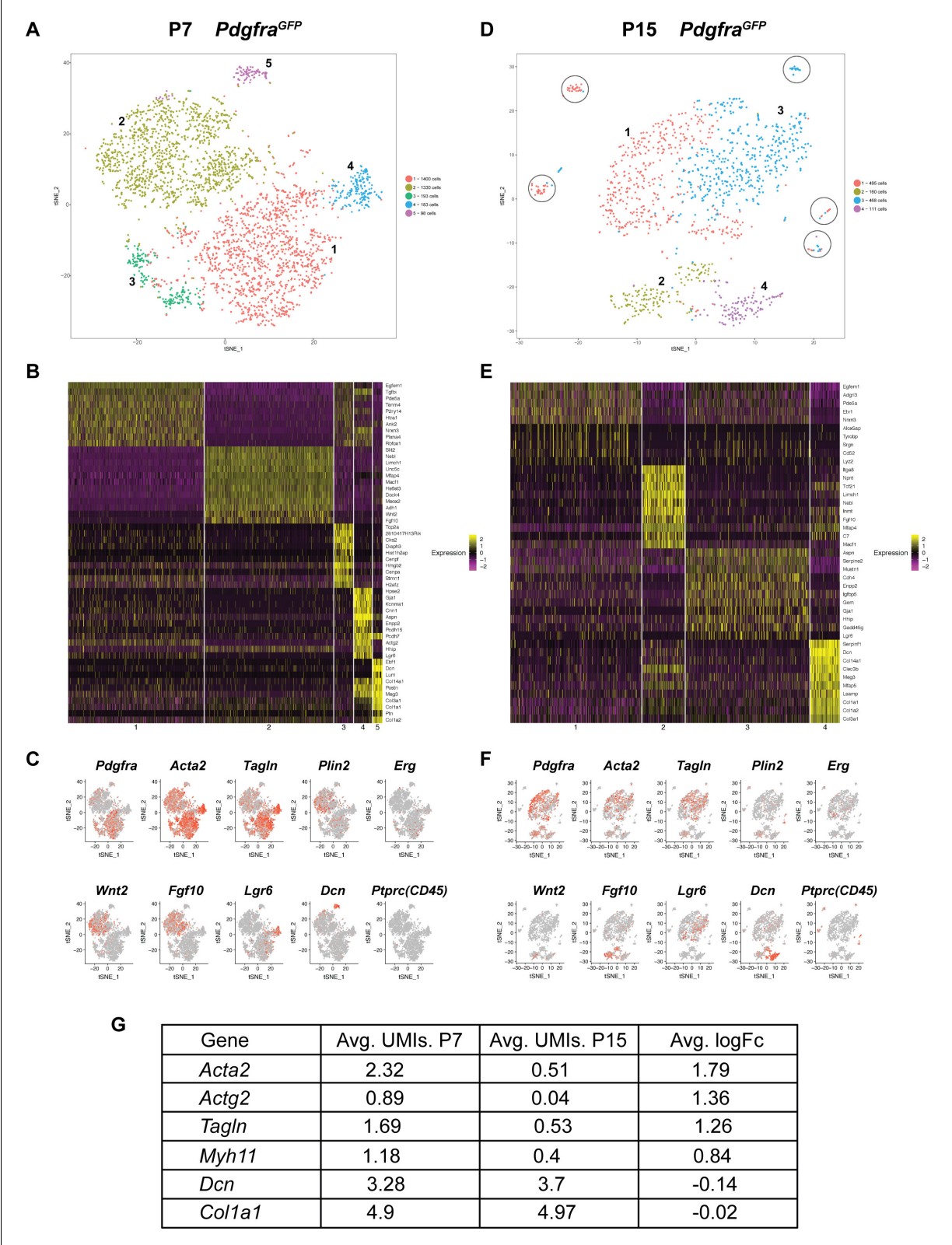

**Figure 3.** Single cell transcriptome analysis of *Pdgfra*+ cells during alveologenesis. (**A, D**) tSNE plots of scRNAseq data from cells isolated from P7 (**A**) and P15 (**D**) *Pdgfra*GFP lungs. Cells can be clustered into five and four main distinct populations at P7 and P15, respectively. Circles in D highlight satellite clusters. (**B, E**) Heatmap showing top expressed genes across clusters from P7 (**B**) and P15 (**E**) *Pdgfra*GFP lungs. (**C, F**) Distribution of cells expressing the indicated marker genes on tSNE plots from P7 (**C**) and P15 (**F**) *Pdgfra*GFP lungs. (**G**) A list of representative genes showed altered

*Figure 3 continued*

expression in P7 versus P15 *Pdgfra*[GFP] lungs, as indicated by the average unique molecular identifier (UMI) of the gene per cell in the *Pdgfra*-high clusters (for *Acta2, Actg2, Tagln* and *Myh11*), or in the matrix clusters (for *Dcn, Col1a1*).

DOI: https://doi.org/10.7554/eLife.36865.012

The following figure supplement is available for figure 3:

**Figure supplement 1.** FACS sorting of GFP+ cells from P15 *Pdgfra*[GFP] mice.

DOI: https://doi.org/10.7554/eLife.36865.013

satellite groups (*Figure 3D*). There are similarities and differences as compared to P7. Cluster one at P15, similar as cluster one at P7, contained a majority of *Pdgfra*-high expressing cells (*Figure 3E,F*). While they still express myofibroblast markers such as *Acta2* and *Tagln*, their level, as well as the level of other myofibroblast-associated genes, are significantly lower at P15 compared to P7 (*Figure 3G*), consistent with our finding from antibody staining. Cluster 2, similar to cluster two at P7, exhibited enriched expression of *Fgf10* and to a less extent *Wnt2* and *Plin2* (*Figure 3E,F*). At P15, we did not detect a cluster equivalent to cluster three at P7, the cluster with cell cycle and cell proliferating gene signature. This is in line with published finding that Ki67, a proliferation marker, correlates with *Pdgfra* expression at P4, but not at P12 (*Kimani et al., 2009*). The P15 cluster three was enriched for cells expressing *Lgr6*, and thereby is similar to cluster four in P7 lungs, with primarily airway smooth muscle cell characteristics (*Figure 3E,F*). The P15 cluster four exhibited enriched expression of matrix associated genes, some of them the same as cluster five markers in P7 lungs, suggesting that there is a small subset of the *Pdgfra*-low population that continued to express matrix-associated genes at a high level (*Figure 3E–G*). At P15 but not at P7, there are several scattered satellite small populations (circled) that are *Pdgfra*-negative, and are positive for *Ptprc* (CD45) and enriched for immune markers (*Figure 3A,C,D,F*). They are likely immune cells that escaped FACS gating.

## Genetic ablation of *Pdgfra*+ cells leads to simplified alveoli

Data from us and others have shown that the *Pdgfra*+ myofibroblasts underline the septal ridges during alveologenesis (*Branchfield et al., 2016*; *Endale et al., 2017*; *McGowan et al., 2008*), raising the hypothesis that *Pdgfra*+ myofibroblasts may be essential for septae formation. To test this directly, we genetically ablated *Pdgfra*+ cells in *Pdgfra*[rtTA];*tetO-cre*;*Rosa-Dta* mice. Neonatal triple transgenic mice received three daily injections of dox at P3-P5 to induce DTA expression at the onset of alveologenesis. The pups were harvested at P15 (*Figure 4A*). After dox induction, qRT-PCR result showed that *Pdgfra*[rtTA];*tetO-cre*;*Rosa-Dta* lungs exhibited a 58% reduction of *Pdgfra* transcripts compared to controls, indicating the efficiency of ablation (*Figure 4B*). This ablation led to a clear emphysema-like enlargement of distal airspaces, a 57% increase as quantified by MLI as compared to control (*Figure 4C,D,I*). This is accompanied by disorganized Elastin, the key extracellular matrix molecule deposited by myofibroblasts (*Figure 4E,F*). *Pdgfra*+ cells have been described as niche for AEC2s by way of supporting their proliferation and differentiation (*Barkauskas et al., 2013*). While there is a decrease of AEC2 cells in the lungs after *Pdgfra*+ cell loss, there is no change in the % AEC2 cells to total, suggesting that the observed AEC2 reduction is proportion to the overall reduction of total cell number due to simplification (*Figure 4G,H and J*). Overall, these findings support the conclusion that the presence of *Pdgfra*+ myofibroblasts are essential for alveologenesis.

## *PDGFRa*-lineaged cells contribute to pathological myofibroblasts in the bleomycin model of lung fibrosis

Multiple studies have shown that PDGF signaling plays an important role in fibrosis. Increased PDGF signaling via transgenic expression of a PDGFA ligand or conditional activation of a gain-of-function PDGFRα mutant form is sufficient to cause tissue fibrosis (*Olson and Soriano, 2009*; *Pontén et al., 2003*). Chemical inhibitors of PDGFRα/β tyrosine kinase activity can reduce lung fibrosis in a radiation induced fibrosis model (*Abdollahi et al., 2005*). To directly test the hypothesis that *Pdgfra* lineaged cells contribute to myofibroblasts in the context of adult pulmonary fibrosis, we carried out lineage tracing in *Pdgfra*[rtTA];*tetO-cre*; *Rosa-tdTomato* triple transgenic mice treated with bleomycin. Mice were given one dose of dox injection one week before bleomycin injury when all alveolar

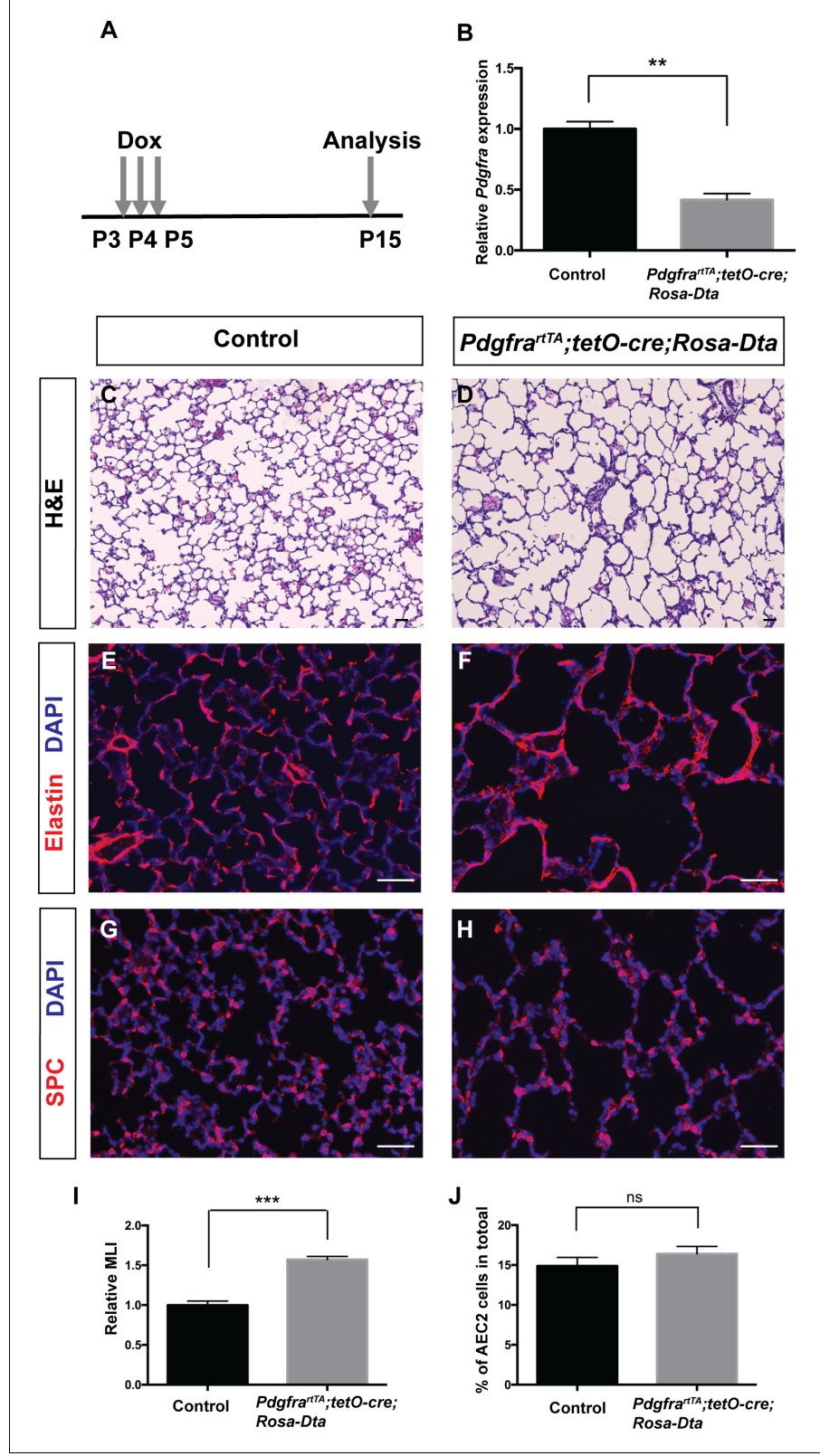

**Figure 4.** Genetic ablation of *Pdgfrα+* cells led to simplified alveoli. (**A**) Timeline of dox treatments and analysis.
(**B**) qRT-PCR analysis indicated decreased *Pdgfra* expression in *Pdgfra^rtTA^;tetO-cre;Rosa-Dta* lungs compared to
controls (1 ± 0.06 for control and 0.42 ± 0.05 for *Pdgfra^rtTA^;tetO-cre;Rosa-Dta* mice, **p=0.0019, n = 3 each). (**C–D**)
Representative H and E stained sections showing that *Pdgfra^rtTA^;tetO-cre;Rosa-Dta* mice have simplified alveoli
*Figure 4 continued on next page*

*Figure 4 continued*

compared to control. (**E–F**) Representative immunofluorescent stained sections for Elastin (red) and DAPI (blue) showing that *Pdgfra*<sup>*rtTA*</sup>*;tetO-cre;Rosa-Dta* lungs have disorganized Elastin. (**G–H**) Representative Immunofluorescent staining for AEC2 cells marker SPC (red) and DAPI (blue) showing that *Pdgfra*<sup>*rtTA*</sup>*;tetO-cre; Rosa-Dta* lungs have reduced AEC2 cells in proportion with alveoli simplification. Scale bars: 50 μm. (**I**) Quantification of alveolar simplification by MLI ($1 \pm 0.05$ for control and $1.57 \pm 0.04$ for *Pdgfra*<sup>*rtTA*</sup>*;tetO-cre;Rosa-Dta* mice, \*\*\*p=0.0009, n = 3 each). (**J**) Quantification of the percentage of total cells that are SPC+. There is no change in the proportion with or without ablation ($14.9 \pm 1.06$ for control and $16.4 \pm 0.93$ for *Pdgfra*<sup>*rtTA*</sup>*;tetO-cre; Rosa-Dta* mice, ns for not significant, p=0.3414, n = 3 each).
DOI: https://doi.org/10.7554/eLife.36865.014
The following source data is available for figure 4:

**Source data 1.** Raw data for *Figure 4*.
DOI: https://doi.org/10.7554/eLife.36865.015

---

*Pdgfra*+ cells were SM22-. Lungs were harvested for histology at either 14 or 21 days after bleomycin administration (*Figure 5A*). Compared to controls treated with PBS at the equivalent stage, we detected a significant increase in lineage-labeled tdTomato+ cells in the bleomycin group (*Figure 5B,E,H,K and N*). A notable subset of these lineaged cells showed proliferative marker Ki67 + in the bleomycin group, but none in the control group (*Figure 5—figure supplement 1*). In addition, there were more lineaged cells that are SM22a+ at 21 days after bleomycin (~45%) than at 14 days (~26%) (*Figure 5B–M and O*). These results suggest that *Pdgfra*-lineaged cells increase in number and upregulate SM22a expression following bleomycin-induced injury, and contribute significantly to pathological myofibroblasts in this model of adult pulmonary fibrosis.

### *Pdgfra*-lineaged cells are decreased following neonatal hyperoxia and are not a major contributor to persisting myofibroblasts

It has been shown that pathological myofibroblasts increase and persist in both human BPD lungs as well as neonatal hyperoxia model of BPD (*Benjamin et al., 2007*; *Bozyk et al., 2012*; *Branchfield et al., 2016*; *Popova et al., 2014*). To determine if *Pdgfra*-lineaged cells contribute to pathological myofibroblasts in the neonatal hyperoxia model of BPD, we examined lineaged cells in *Pdgfra*<sup>*rtTA*</sup>*;tetO-cre;Rosa-tdTomato* triple transgenic mice following treatment. Pregnant females were fed dox food until P0, when the females and their pups were switched to normal food that do not contain dox, and raised in 75% O2 until analysis at P12 (*Figure 6A*). As expected, compared to room air controls, exposure to 75% O2 resulted in simplified alveoli, and a persistence of high SM22a staining at P12 when SM22a staining is already low in room air control lungs (*Figure 6B–G*). In contrast to the increase of lineaged cells following bleomycin injury, neonatal hyperoxia led to a statistically significant decrease in the tdTomato+ *Pdgfra*-lineaged cells (*Figure 6B–D*). As a consequence, few of the high SM22a+ cells showed tdTomato lineage marker (*Figure 6F*). A previous in vitro study showed that hyperoxia treatment of cultured lung mesenchymal cells lead to an increase of myofibroblasts and concomitant decrease of lipofibroblasts (*Rehan and Torday, 2003*). Consistent with this, we found a statistically significant decrease of lipofibroblast marker ADRP in neonatal hyperoxia treated lungs (*Figure 6H–J*), raising the possibility that a possible lipofibroblasts to myofibroblasts transdifferentiation may account for the decrease in the former and the increase in the latter cell type. Taken together, our findings indicate that *Pdgfra*-lineaged cells contribute significantly to pathological myofibroblasts in the bleomycin model of lung fibrosis, but not in the neonatal hyperoxia model of BPD.

## Discussion

To address the origin and plasticity of tissue mesenchymal cell types, we focused on *Pdgfra*, which has been extensively implicated as a marker of progenitors and/or progenitor niches in normal development and disease pathogenesis (*Green et al., 2016*; *McGowan and McCoy, 2014*; *Rock et al., 2011*; *Zepp et al., 2017*). Using the novel knockin *Pdgfra*<sup>*rtTA*</sup> line, we were able to precisely capture *Pdgfra* expression and rigorously interrogate the in vivo lineage of these cells. Our

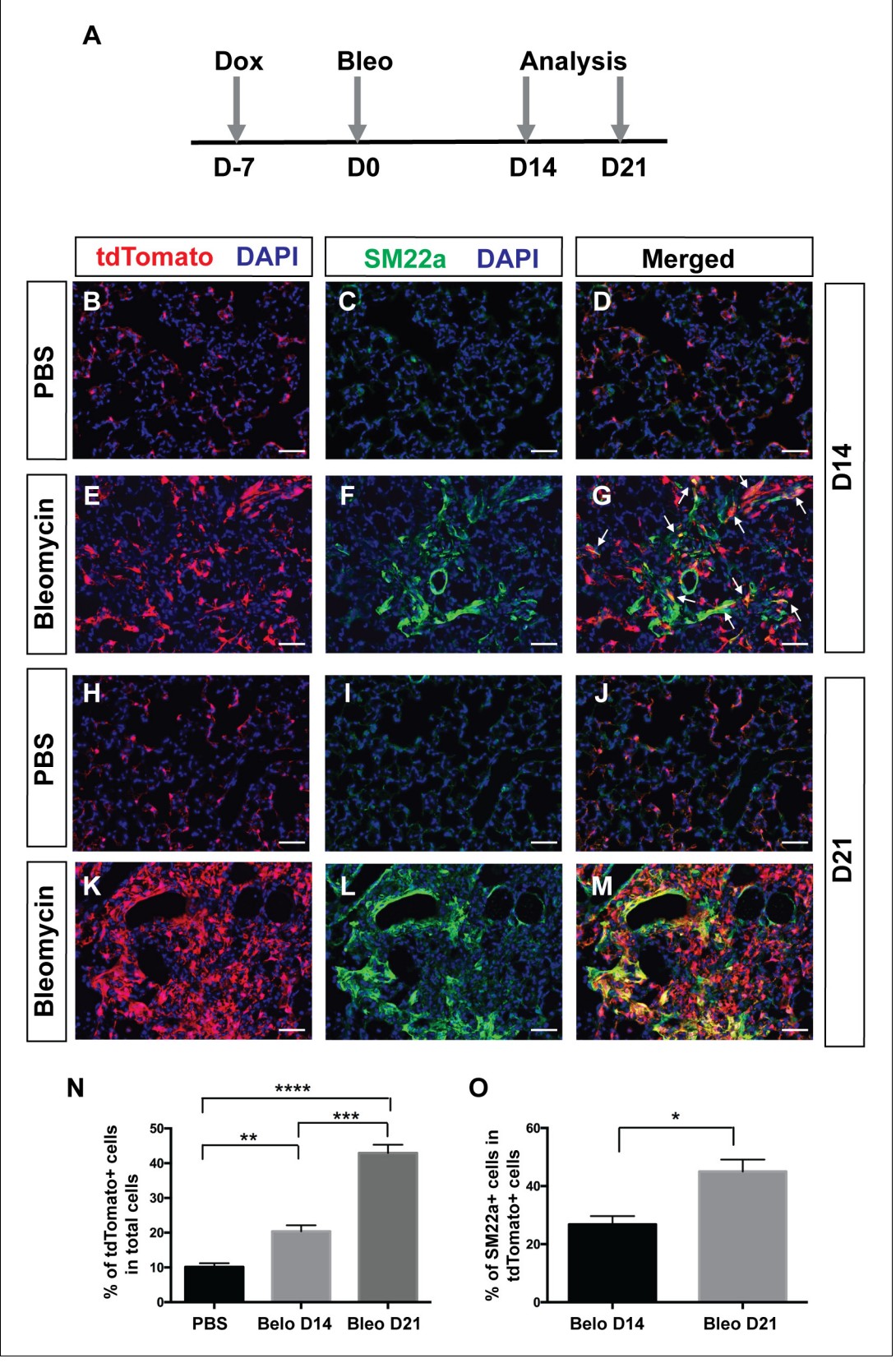

**Figure 5.** *Pdgfra*-lineage cells increased and contributed to myofibroblasts following bleomycin-induced injury. (A) Timeline of dox treatment, bleomycin administration and analysis. *Pdgfra^{rtTA};tetO-cre;Rosa-tdTomato* mice were
*Figure 5 continued on next page*

*Figure 5 continued*

given one dose of dox injection 7 days (**D**) before bleomycin injury. Lungs were analyzed at either D14 or D21 after bleomycin administration. (**B–M**) Representative immunofluorescent staining for tdTomato (red), SM22a (green) and DAPI (blue) in lungs of PBS-treated controls at D14 (**B–D**), bleomycin-treated mice at D14 (**E–G**), PBS-treated controls at D21 (**H–J**), and bleomycin-treated mice at D21 (**K–M**). (**N**) Quantification of the mean percentage of tdTomato+ cells in total (10.14 ± 1.08 for PBS, 20.4 ± 1.75 for bleomycin D14 and 42.93 ± 2.39 for bleomycin D21. PBS versus bleomycin D14, **p=0.0025; PBS versus bleomycin D21, ****p=0.000016; bleomycin D14 versus bleomycin D21, ***p=0.0003; $n$ = 4 each). (**O**) Quantification of the mean percentage of SM22a + cells in tdTomato+ cells (26.85 ± 2.85 for bleomycin D14 and 45.01 ± 4.1 for bleomycin D21, *p=0.0109, $n$ = 4 each). Scale bars: 50 μm.

DOI: https://doi.org/10.7554/eLife.36865.016

The following source data and figure supplement are available for figure 5:

**Source data 1.** Raw data for *Figure 5*.
DOI: https://doi.org/10.7554/eLife.36865.018
**Figure supplement 1.** Pdgfra-lineaged cells proliferate following bleomycin treatment.
DOI: https://doi.org/10.7554/eLife.36865.017

findings using this line revealed that these cells show distinct potential in giving rise to the different types of myofibroblasts through development and pathogenesis (*Figure 7*).

Lineage tracing during development showed that shortly after birth as the lung enters into the alveologenesis program to build new gas-exchange surface area (P3-P14), *Pdgfra*+ cells primarily (~95%) give rise to myofibroblasts, with minor (~5%) contribution to lipofibroblasts. This differs from the results from a recent lineage tracing study using a *Pdgfra-CreERT2* BAC transgenic line, which showed that lineaged cells gave rise to more lipofibroblasts than myofibroblasts at the same stage (P7) (*Ntokou et al., 2015*). The difference of the results may be due to differences in the cell populations that are captured by the knock-in line versus the transgenic line. Despite being a BAC transgenic line, it may not recapitulate endogenous *Pdgfra* expression because it may be missing some required remote regulatory sequences, or it may be influenced by enhancers near the random transgenic insertion site. Conversely, the difference could be due to that our *Pdgfra*$^{rtTA}$ knockin line is missing one copy of *Pdgfra* and thereby deficient in some aspects of lineage contribution. However, there is no notable phenotype in the *Pdgfra* heterozygotes to indicate haploinsufficiency. In addition, conditional *Pdgfra* loss-of-function mutants show decreased myofibroblasts and maintenance of lipofibroblasts (*McGowan and McCoy, 2014*). Thus, if there is any undetected happloinsufficiency in our *Pdgfra*$^{rtTA}$ knockin line, the prediction would be a lower lineage contribution to myofibroblasts compared to the BAC transgenic line, opposite to the data shown. Interestingly, ~4% of *Pdgfra*+ cells give rise to pulmonary endothelial cells in prenatal tracing but not in postnatal tracing. This is consistent with previous result showing that lung endothelial cells can be labeled by *Tbx4-rtTA* lineage tracing in the early stages but not late stages of development (*Zhang et al., 2013*).

Using our knockin line, lineage tracing into the early adult at P40 showed that *Pdgfra*-lineaged cells persist. However, they no longer express myofibroblast markers, since a vast majority of lung mesenchymal cells has turned down myofibroblast and lipofibroblast marker expression. Short of the identification of a differentiation marker that distinguished them from others, it appears that the *Pdgfra*+ cells may have de-differentiated to a non-descriptive, resting mesenchymal cell state. This dynamic change of cellular characteristics is distinct from the behavior of epithelial cells, where differentiated cells rarely de-differentiate unless following severe injury (*Pardo-Saganta et al., 2015*).

Our scRNAseq data are consistent with lineage tracing data, and further add to the resolution of sub-populations within *Pdgfra*+ cells. At P7, a majority of the *Pdgfra*+ cells expressed myofibroblast markers while a minority expressed lipofibroblast marker. Cluster 1, with higher density of *Pdgfra* + cells, are more enriched for myofibroblast markers. In comparison, cluster 2, with lower density of *Pdgfra*+ cells, are more enriched for lipofibroblast markers. This result is consistent with prior finding from the McGowan group showing that the *Pdgfra*-high cells have myofibroblast characteristics, while the Pdgfra-low cells have lipofibroblast characteristics. Comparing P15 to P7, the cluster with cell proliferation signature disappeared and the expression of myofibroblast markers are much reduced, consistent with prior findings (*Branchfield et al., 2016*; *Kimani et al., 2009*). One cluster that persisted at both stages is enriched for matrix-associated genes. It remains to be determined

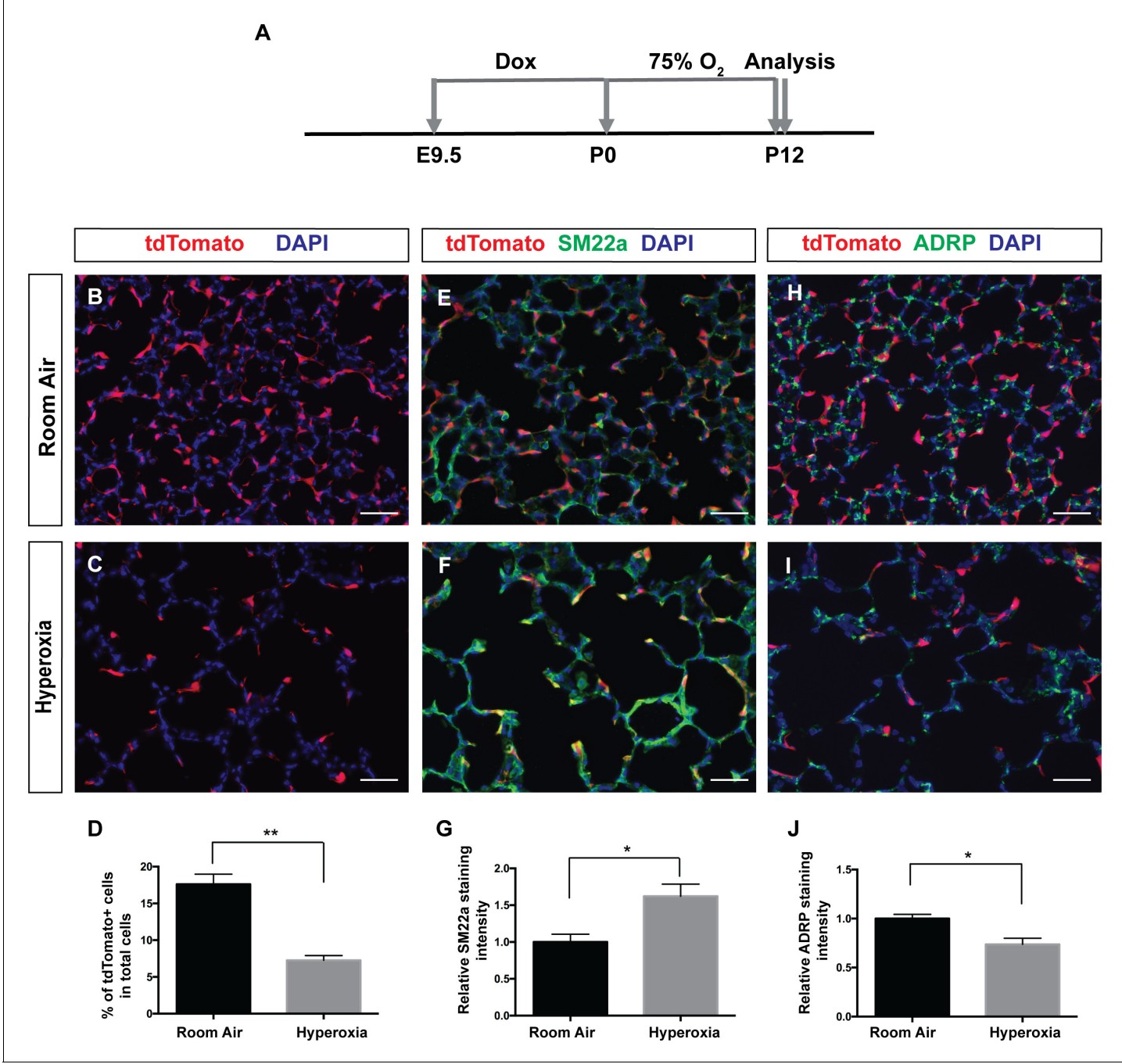

**Figure 6.** *Pdgfra*-lineaged cells decrease following neonatal hyperoxia-induced injury. (A) Timeline of dox and hyperoxia treatment. Pregnant females were fed dox food between E9.5 and P0. Pups and mother were then fed food without dox, raised in 75% O2 or room air from P0 to P12 and the lungs were harvested at P12. (B, C) Representative immunofluorescent staining for tdTomato (red) and DAPI (blue) showing that the hyperoxia group have decreased tdTomato-lineaged cells compared to room air controls. (D) Quantification showing the percentage of tdTomato+ cells in room air and hyperoxia (17.6 ± 1.37 for room air and 7.25 ± 0.65 for hyperoxia, \*\*p=0.0024, n = 3 each). (E, F) Representative immunofluorescent staining for tdTomato (red), SM22a (green) and DAPI (blue) showing that the hyperoxia group have increased SM22a+ myofibroblasts. (G) Quantification showing relative SM22a staining intensity as quantified in relationship to DAPI intensity (1 ± 0.11 for room air and 1.62 ± 0.17 for hyperoxia, \*p=0.0342, n = 3 each). (H, I) Representative immunofluorescent staining for tdTomato (red), ADRP (green) and DAPI (blue) showing that the hyperoxia group have decreased ADRP+ lipofibroblasts. (J) Quantification showing relative ADRP staining intensity as quantified in relationship to DAPI intensity (1 ± 0.04 for room air and 0.74 ± 0.06 for hyperoxia, \*p=0.0257, n = 3 each). Scale bars: 50 μm.
DOI: https://doi.org/10.7554/eLife.36865.019

The following source data is available for figure 6:

*Figure 6 continued on next page*

*Figure 6 continued*

**Source data 1.** Raw data for *Figure 6*.
DOI: https://doi.org/10.7554/eLife.36865.020

how this population compares to the previously described matrix-fibroblasts from the adult lung, as distinct matrix markers are displayed (*Green et al., 2016*). Regardless, it is of interest that during the two phases of alveologenesis as represented by P7 and P15, there remains a small *Pdgfra+* subbpopulation that shows enriched matrix gene expression.

Data from the fibrosis model indicate that the adult *Pdgfra$^+$* cells, while having lost their myofibroblast marker expression, are still capable of becoming myofibroblasts by turning on SM22a expression under injury. They also retained the ability to proliferate, which led to population expansion after injury (*Rock et al., 2011*; *Zepp et al., 2017*). These *Pdgfra*-lineaged cells are not the only source of pathological myofibroblasts in fibrosis. For example, a recent study showed that *Plin2*-lineaged lipofibroblasts can also give rise to a-SMA+ cells in fibrosis (*El Agha et al., 2017*).

In contrast to the fibrosis model, in the neonatal hyperoxia model of BPD, our data showed that the *Pdgfra*-lineaged cells do not contribute significantly to myofibroblasts. This is unexpected as neonatal hyperoxia treatment occurred at the early postnatal stage when the *Pdgfra+* cells normally contribute to myofibroblasts. Our findings showed that under hyperoxia, however, a majority of lineaged cells were lost, presumably through cell death, thereby unable to contribute to myofibroblasts. Yet, other cells differentiate into myofibroblasts. The result that lipofibroblasts were reduced in number, coupled with in vitro data that hyperoxia can drive cultured lung mesenchymal cells to preferentially adopt myofibroblast fate instead of lipofibroblast fate, raised the possibility that the lipofibroblasts may transdifferentiate into myofibroblasts in the neonatal hyperoxia model (*Rehan and Torday, 2003*). This possibility remains to be tested through lineage tracing of lipofibroblasts. Recently, a *Plin2* (*Adrp*)$^{creERT2}$ line was generated to label lipofibroblasts (*El Agha et al., 2017*; *Ntokou et al., 2017*). However, it also labels a number of other cell types in the lung (*El Agha et al., 2017*). Thus, generation of a genetic tool that specifically labels lipofibroblasts will be critical to address the potential of these cells in development and injury.

Considering the three types of myofibroblasts in lung: the normal secondary crest myofibroblasts, adult fibrotic myofibroblasts, and BPD myofibroblasts, our data substantiate the notion that not all myofibroblasts are created equal. In terms of their origins, they have different contributions from

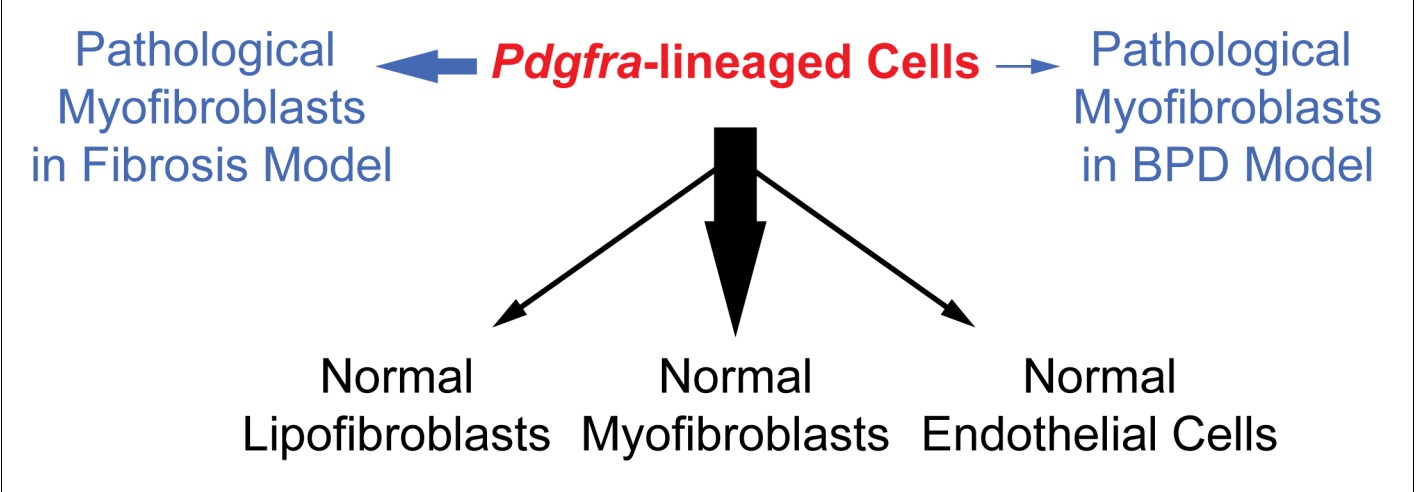

**Figure 7.** A model illustrating the lineage potential of *Pdgfra*-lineaged cells in development and pathogenesis. In normal development (black notes), *Pdgfra*-lineaged cells give rise primarily to myofibroblasts, and minimally to lipofibroblasts and endothelial cells. In pathogenesis (blue notes), *Pdgfra*-lineaged cells contribute substantially to myofibroblasts in the bleomycin model of fibrosis, but contribute minimally to myofibroblasts in the neonatal hyperoxia model of BPD. Black and blue arrows are weighted to indicate distinct extent of contribution.
DOI: https://doi.org/10.7554/eLife.36865.021

*Pdgfra*-lineaged cells. Furthermore, our data support that the smooth muscle marker positive state may be a transient characteristic rather than a stable cell fate (*Figure 7*). This transient nature is not only a feature in development, but it also occurs in disease settings. For example, in fibrosis model, it has been shown that a-SMA+ cells can revert to lipofibroblasts, which contributes to resolution of the disease (*El Agha et al., 2017*).

A number of recent studies have used transcriptome, lineage tracing and epithelium-mesenchyme organoid approaches to determine the complexity of the lung mesenchyme (*Endale et al., 2017*; *Lechner et al., 2017*; *Lee et al., 2017*; *Li et al., 2015*; *Zepp et al., 2017*). For examples, genes that are WNT reporters such as *Lgr5*, *Lgr6*, and *Axin2*, or HH reporter such as *Gli1* have been used to lineage trace cell fate in the mesenchyme in both baseline as well as in injury settings. A next challenge is to compare the different lineaged populations to define the similarities and differences, and be able to comprehensively illustrate the complexity of the mesenchyme.

# Materials and methods

**Key resources table**

| REAGENT or RESOURCE | Designation | Source | Identifier | Additional information |
|---|---|---|---|---|
| strain, strain background (mouse) | *tetO-GFP*, mix background | Jackson Laboratories | 016836 | |
| strain, strain background (mouse) | *Pdgfra*<sup>GFP</sup>, mix background | Jackson Laboratories | 007669 | |
| strain, strain background (mouse) | *tetO-Cre*, B6 background | Jackson Laboratories | 006234 | |
| strain, strain background (mouse) | *Rosa-tdTomato*, B6 background | Jackson Laboratories | 007914 | |
| strain, strain background (mouse) | *Rosa-Dta*, mix background | Jackson Laboratories | 006331 | |
| Antibody | Elastin | Abcam | ab21600, 1:500 | |
| Antibody | SM22a | Abcam | ab14106, 1:200 | |
| Antibody | ADRP | Abcam | ab52356, 1:200 | |
| Antibody | ERG | Abcam | ab92513, 1:200 | |
| Antibody | ICAM2 | BD | 553325, 1:200 | |
| Antibody | Ki67 | Abcam | ab15580, 1:200 | |
| Antibody | a-SMA | Sigma | F3777, 1:200 | |
| Antibody | goat anti rabbit FITC | Jackson Immuno Research Laboratories | 111-095-144, 1:200 | |
| Antibody | goat anti rabbit Cy3 | Jackson Immuno Research Laboratories | 111-165-144, 1:200 | |
| Antibody | goat anti rat FITC | Jackson Immuno Research Laboratories | 112-095-003 1:200 | |

## Mice

To generate *Pdgfra*<sup>rtTA</sup> knockin mouse strain, 5'-GAGACAGTAATTGAGGATCCTGG-3' was designed to target a PAM site in the first intron of the *Pdgfra* gene. A plasmid donor was

constructed by ligating the splice acceptor (SA), rtTA, and a trimerized simian virus 40 polyadenylation (3 pA) (*Hamilton et al., 2003*; *Zhang et al., 2013*). The homologous arms (1.5 kb each) were amplified from C57BL/6 mouse genomic DNA and added. *Pdgfra*$^{rtTA}$ mouse line was generated at the University of Wisconsin-Madison transgenic animal facility by injection of 50 ng/μl gRNA, 40 ng/μl CAS9 protein and 25 ng/μl donor plasmid. The founders were identified by PCR using primers: LF, 5'-AGCAACTAACTAAAGCATGGTC-3'; LB, 5'-GCTCAACTCCCAGCTTTTG-3'; RF, 5'-GAAA TTGCATCGCATTGTCTG-3'; RB, 5'-ACTCTCATCCGTCTGAGTG-3'. Mice were housed and all experimental procedures were performed in American Association for Accreditation of Laboratory Animal Care-accredited laboratory animal facilities at the University of Wisconsin and University of California San Diego. Prenatal lung *Pdgfra*$^{rtTA}$ activity was induced by doxycycline (dox) administration starting at different gestational periods by feeding pregnant females with dox food (625 mg/kg; Test- Diet, Richmond, IN, USA). In postnatal pups and adult mice, *Pdgfra*$^{rtTA}$ activity was induced by intraperitoneal injection of dox (100 mg/kg body weight). For prenatal lineage tracing, we chose dox containing food because it is gentle on the pregnant females and effective at inducing activity. For postnatal lineage tracing, we chose dox injection followed by dox food because it showed a much higher efficiency compared to dox containing food only, presumably because the pups do not get to the dox food themselves and the dox transfer through the milk from the dam is not efficient.

## Neonatal hyperoxia model of BPD

Hyperoxia-exposure was conducted as previously described (*Branchfield et al., 2016*). Briefly, at P0, new born pups with their mother were placed into a chamber (BioSpherix) with circulated 75% oxygen through to P12. Control pups born at the same time were maintained at normoxia (room air) conditions for the same duration of the experiment. Mothers were swapped between normoxia and hyperoxia litters every fourth day to maintain maternal health. Lungs from both normoxia and hyperoxia groups were harvested at P12.

## Bleomycin model of adult lung fibrosis

Mice were anesthetized with ketamine (100 mg/kg) and xylazine (15 mg/kg) prior to delivering a single dose of intratracheal bleomycin (1 unit/kg, Teva Pharmaceutical Industries) dissolved in 50 μL of 0.9% NaCl Irrigation (Abbott Laboratories) via endotracheal intubation. Daily weight and wellness checks were performed until euthanasia at 14 or 21 days post-bleomycin.

## Histology and immunofluorescence staining

For histological analysis, postnatal mouse lungs were gravity inflated and fixed overnight with 4% paraformaldehyde, and submerged in 30% sucrose at 4°C overnight. Whole lung lobes were embedded in OCT compound (Sakura) for cryo section. Sections were stained using a standard Hematoxylin and Eosin (H and E) protocol. Quantification of lung simplification was performed using the mean linear intercept (MLI) method as previously described (*Li et al., 2017*).

Immunofluorescence staining was performed following standard protocols and mounted with VECTASHIELD medium with DAPI (Vector Laboratories). Antibodies were listed in the resource table.

## Quantitative RT-PCR

Total RNA was extracted from lungs using Trizol (Invitrogen) and an RNeasy Micro RNA extraction kit (Qiagen). RNA was reverse transcribed using the iScript Select cDNA Synthesis Kit (Bio-Rad). Quantitative PCR was performed using SYBR Green (Bio-Rad). Three technical and three biological replicates were performed for each gene. Primers used for qPCR are: *β-actin*-F: 5'-CGGCCAGGTCA TCACTATTGGCAAC-3'; *β-actin*-R: 5'-GCCACAGGATTCCA-TACCCAAGAAG-3'; *Pdgfra*-F: 5'-TGCGGGTGGACTCTGATAATGC-3'; *Pdgfra*-R: 5'-GTGGAACTACTGGAACCTGTCTCG-3'.

## Lung dissociation, flow cytometry and Single-Cell RNA-sequencing

Lung lobes were separated and minced into small pieces in a conical tube containing 6 ml of 2 U/ml dispase (STEMCELL Technologies) followed by rotating incubation for 30 min at 37°C. 15 μl 10 mg/ml DNase I (Sigma) was added and cells were incubated on ice for 5 min. Cells were filtered through 40 μm strainers and centrifuged at 800 rpm for 6 min at 4°C. Cell pellet was resuspended in 1 ml of

RBC lysis buffer (Abcam) and lysed for 90 s at room temperature. This was followed by slow and gentle addition of 6 ml DMEM media (GIBCO) and 500 mL of FBS (Hyclone). Cells were centrifuged at 800 rpm for 6 min at 4°C. Cell pellet was resuspended in PBS with 3% BSA (GIBCO) for further staining using CD140a (PDGFRa)-PE (eBioscience, 12-1401-81, 1:100) for flow cytometry. 4′, 6-diami-dino-2-phenylindole (DAPI) (Sigma) was used to eliminate dead cells. Flow sorting was performed with a FACS Aria II (BD Biosciences) and data were analyzed with FlowJo software (Tree Star, Inc.).

For Single-Cell RNA-sequencing, $Pdgfra^{GFP}$ mouse lungs at P7 and P15 were harvested and single cell preparations were FACS sorted as described above. The sorted cells were then processed following 10X genomics protocol. The sequencing libraries were validated and sequenced using HiSeq4000 platform. Sequencing reads were first aligned to mm10 reference using STAR V2.5.1b (*Dobin et al., 2013*) and then aligned to annotated transcripts, which were part of the Cell Ranger version 2.1 pipeline (https://www.10xgenomics.com). A customized pre-mRNA reference package was generated to capture all the intronic reads in addition to reads that are mapped to exons. Cell Ranger filtered cells by taking >10% of the top $n^{th}$ barcodes, where n is 1% of the expected number of cell counts being recovered (*Zheng et al., 2017*). The filtered gene-cell matrices, output from CellRanger, was further analyzed using the Seurat package V2.3 (*Butler et al., 2018*) and R 3.4 (*Team, 2014*). The UMI raw counts were log-normalized for each cell to its total expression and scaled to z-scored residuals of linear models that predicts gene expression using normalized UMIs. The scaled data were used to perform dimensionality reduction with Sparsesvd V0.1 (*Berry, 1992*), graph-based clustering and visualization with t-Distributed Stochastic Neighbor Embedding (t-SNE) (*van der Maaten, 2014*). Differential gene expression analysis was performed using Wilcoxon rank sum test. Heatmaps, t-SNE plots and visualization of marker expression were generated using Seurat. GEO accession number for the scRNA-seq data is GSE118555.

## Quantification and statistical analysis

For quantification of the number of immunofluorescence stained cells, or immunofluorescence staining intensity per field, four independent 20x fields per sample were analyzed. Sections from at least three lungs were quantified for each experimental group. $P$-values were calculated with two-tailed unpaired Student's $t$-test on Prism six software (GraphPad). Data were presented as mean ± SEM, and results were considered statistically significant if $p < 0.05$.

## Acknowledgements

We thank Dr. Philippe Soriano at Icahn School of Medicine at Mt. Sinai and Dr. Wei Shi at Children's Hospital Los Angeles for DNA constructs used in the generation of $Pdgfra^{rtTA}$ donor plasmid. We thank all the members of the XS laboratory, and Dr. James Hagood for constructive discussions. This work was supported by National Heart, Lung, and Blood Institute grants (RO1 HL142215, RO1 HL097134, RO1 HL122406) and a UCSD Pediatrics pilot funding to XS.

## Additional information

### Funding

| Funder | Grant reference number | Author |
|---|---|---|
| National Heart, Lung, and Blood Institute | RO1 HL142215,RO1 HL097134,RO1 HL122406 | Xin Sun |
| UCSD Pediatrics pilot funding | | Xin Sun |

The funders had no role in study design, data collection and interpretation, or the decision to submit the work for publication.

### Author contributions

Rongbo Li, Data curation, Investigation, Writing—original draft, Writing—review and editing; Ksenija Bernau, Nathan Sandbo, Resources, Writing—review and editing; Jing Gu, Sebastian Preissl, Data curation, Writing—review and editing; Xin Sun, Data curation, Supervision, Funding acquisition, Writing—review and editing

## Author ORCIDs
Rongbo Li (iD) http://orcid.org/0000-0001-9112-9712
Xin Sun (iD) https://orcid.org/0000-0001-8387-4966

## Ethics
Animal experimentation: This study was performed in strict accordance with the recommendations in the Guide for the Care and Use of Laboratory Animals of the National Institutes of Health. All of the animals were handled according to approved institutional animal care and use committee (IACUC) protocols (S16187) of the University of California at San Diego.

## Decision letter and Author response
Decision letter https://doi.org/10.7554/eLife.36865.026
Author response https://doi.org/10.7554/eLife.36865.027

## Additional files

### Supplementary files
• Transparent reporting form
DOI: https://doi.org/10.7554/eLife.36865.022

### Data availability
All data generated or analyzed during this study are included in the manuscript and supporting files. Source data files have been provided for Figures 1,2,4-6, Figure 1-figure supplement 1 and Figure 2-figure supplement 2.

The following dataset was generated:

| Author(s) | Year | Dataset title | Dataset URL | Database, license, and accessibility information |
|---|---|---|---|---|
| Li R, Sun X | 2018 | The single cell RNA seq of PDGFRa-GFP+ cells in mouse lung | https://www.ncbi.nlm.nih.gov/geo/query/acc.cgi?acc=GSE118555 | Publicly available at the NCBI Gene Expression Omnibus (GSE118555) |

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
