## [Decision Letter]

Thank you for submitting your article "*Pdgfra* Marks a Cellular Lineage with Dynamic Contributions to Myofibroblasts in Lung Maturation and Injury Response" for consideration by *eLife*. Your article has been reviewed by three peer reviewers, one of whom is a member of our Board of Reviewing Editors, and the evaluation has been overseen by Harry Dietz as the Senior Editor. The reviewers have opted to remain anonymous.

The reviewers have discussed the reviews with one another and the Reviewing Editor has drafted this decision to help you prepare a revised submission.

Your work carefully describes the generation and initial characterization of a new *Pdgfra^rtTA^* mouse line which you have used to assess lineage relationships in the developing and adult mouse lung. All three reviewers felt that your study had merit but noted important weakness that required additional experimental clarification. Three major issues were noted by the reviewers: 1) a better characterization of the *Pdgfra^rtTA^* mouse line as it pertains to fidelity of endogenous *Pdgfra* expression during lung development and in the adult is required and should include quantification of the number of *Pdgfra* expressing cells at the stages examined; 2) the relationship between *Pdgfra* expressing cells, as noted by the *Pdgfra^rtTA^* mouse line, and other mesenchymal populations including SM22a expressing myofibroblasts and ADRP expressing fibroblasts is important to define with additional experimentation; and 3) the need to use additional markers for myofibroblasts beyond SM22a given the variable definition of this cell type. Additional review points should also be addressed as noted.

Reviewer #1:

This report describes the generation of a new *Pdgfra^rtTA^* line which the authors have used to perform cell lineage tracing experiments in mice. The *Pdgfra^rtTA^* mice appear to mark a similar (although not proven) subset of cells that the *Pdgfra^GFP^* knockin mice do. During development, *Pdgfra^rtTA^* mice mark a significant proportion of lung mesenchyme and depletion of *Pdgfra+* cells during alveologenesis causes an emphysema like phenotype. Hyperoxia injury causes a decrease in the number of embryonically *Pdgfra+* lineage traced cells. Conversely, bleomycin induced injury in adult animals labels a significant number of *Pdgfra+* cells that become myofibroblasts.

The paper is well written and the generation of this new animal model is an important contribution to the field. However, there are a number of questions regarding the lineage tracing studies and their interpretation:

1) Since this is the first report of this new mouse model, additional characterization of its fidelity is warranted at each of the induction timepoints. The co-expression of the rtTA/GFP from this allele needs to be looked at more carefully including FACS analysis and additional immunohistochemistry to verify what% of total *Pdgfra+* and other cell types it marks.

2) The lineage tracing studies are interesting but not well defined. For instance, the labeling of *Pdgfra+* cells at E9.5 could simply mark almost all lung mesenchyme/fibroblasts. If this is the case, then the interpretation of data in Figures 2 and 5 are somewhat unclear. The authors should provide an index at all of the times of dox induction (i.e. E9.5, P3, adult etc.) of what% of the total lung mesenchyme this allele marks. A comparison to a pan-mesoderm or mesenchyme marker would be helpful i.e. vimentin or something similar.

3) In light of point 2 above, what do the authors make of the fact that SM22a and ADRP appear to mark the majority of alveolar mesenchyme at P7 (Figure 2)? In particular, ADRP seems to mark a lot of cells that are *Pdgfra-*. A better characterization of what these cells are is warranted.

4) The data in Figure 5 and Figure 5—figure supplement 1 are interesting in that they show a significant number of *Pdgfra+* cells that proliferate in response to bleomycin injury. Are these the myofibroblasts or are they what appear to be a large number of SM22-/*Pdgfra+* lineage traced cells?

5) The IHC data shown in Figure 5 appear to show that SM22+ cells at day 14 are unlikely to be derived from *Pdgfra-*lineage. Is this due to early vs late contribution to bleomycin triggered myofibroblasts?

6) The cell ablation experiment in Figure 3 suggests that the loss of *Pdgfra-*derived early postnatal myofibroblasts results in alveolar simplification. While the authors note that AT2 cell numbers are reduced, the percentage to total lung cells is unchanged. It is unclear how the latter conclusion was made or whether total cell number or epithelial cells specifically, were altered.

Reviewer #2:

Here the authors investigate the lineage potential of *Pdgfra-*expressing cells in the developing lung and the contribution of this lineage to lung injury-repair in different well-established models. By generating and analyzing a novel *Pdgfra^rtTA^;tetO-cre* knockin reporter mouse, they report that *Pdgfra-*expressing cells give rise largely to myofibroblasts with only modest generation of lipofibroblasts in the lung mesenchyme. Moreover, by using these mice to induce DTA-expression and ablate myofibroblasts postnatally, or to injure the adult lung with bleomycin, the authors demonstrate the key role of *Pdgfra-*lineage in alveolar formation and in mediating the fibrotic response, respectively. Lastly, they report no contribution of this lineage to the persistent myofibroblast population seen in neonatal hyperoxic lung injury.

This interesting and succinct paper provides both confirmatory and new insights about the fate of lineage-derived *Pdgra+* cells in different settings. The study is carefully done and relies on a more accurate knockin mouse model to investigate the *Pdgfra* lineage, which challenges some of the results previously reported using a BAC transgenic line.

Given the novelty of this mouse model and its potential importance to address relevant questions in the field, the details about the characterization of this line are relatively brief and sometimes missing. For example, they use two different regimens of Dox administration (IP and dox-containing food) but do not comment on the rationale and whether there were differences in labeling efficiency. What was the frequency of dox-independent labeling in these mice? Is labeling present in smooth muscle elsewhere in airways or large vessels? As a knockin line, *Pdgfra^rtTA^;tetO-cre* mice are haploinsufficient for *Pdgfra*. The authors should provide evidence or comment whether this may have influenced any of the results reported here, including the differences in labeling from that reported in the BAC transgene line.

The surprising and apparently paradoxical high number of myofibroblasts *Pdgfra*-lineage labeled negative in the BPD-hyperoxia model is one of the most interesting observations in the paper. The conclusion that this may have been accounted for by lipofibroblasts transdifferentiating into myofibroblasts should be explored in this model by lineage-labeling lipofibroblasts (ADRP or other relevant cre line). This will strengthen their conclusion and will offer further support to their *Pdgfra^rtTA^;tetO-cre* mouse model.

Reviewer #3:

This manuscript by Li et al. describes the generation of a new line of mice in which rtTA is placed under the regulatory control of the endogenous *Pdgfra* promoter. This new line was validated by comparing specificity of Dox-dependent *tetO-GFP* expression with expression of a GFP reporter in lungs of *Pdgfra^GFP^* mice. Authors go on to show that *Pdgfra*-lineage cells of the embryonic lung transiently generate SM22a-immunoreactive cells in the perinatal period which then contribute to lipofibroblasts, that ablation of *Pdgfra*-expressing cells in the perinatal period leads to alveolar simplification and that *Pdgfra*-lineage cells differentially contribute to SM22a-immunoreactive cells in models of neonatal hyperoxia and adult bleomycin-induced fibrosis. Strengths of this study are the demonstration of a valuable new genetic tool to investigate the biology of *Pdgfra*-expressing cells of the lung and application of this tool to study the dynamics of lung mesoderm during development and perinatal lung maturation. A concern with this study is the broad classification of SM22a+ cells of the developing, perinatal and adult lung as myofibroblasts. Lineage tracing of *Pdgfra*-expressing cells is performed at various times of embryonic or postnatal lung maturation which complicates interpretation of data since *Pdgfra* is dynamically expressed within mesenchymal cells during these stages of lung development/maturation.

Specific concerns include:

1) The conclusion that ".hitherto unappreciated complexity in the behavior of the *Pdgfra*-lineaged cells" seems overstated. Previous studies by this group and others demonstrate that the molecular phenotype of lung mesoderm is dynamically regulated during development.

2) Results subsection “Lineage tracing revealed that Pdgfra cells give rise primarily to normal alveolar myofibroblasts”: “[…] demonstrate that in the early postnatal lung, *Pdgfra*-lineage cells give rise primarily to myofibroblasts, with minor contributions to lipofibroblasts and endothelial cells.” This statement is a little misleading in that it implies that myofibroblast and lipofibroblast populations are distinct and fixed. In reality, as shown previously by this group (Branchfield et al., 2016) and nicely demonstrated in this study, these populations are dynamic and are either interchangeable or their relative populations change through "pruning" during early postnatal lung maturation. The authors provide evidence supporting the notion that these populations are interchangeable based upon long-term retention of lineage-labeled cells. A concern with these experiments is that all conclusions are made using immunofluorescence analysis to evaluate fibroblast phenotype in histologic sections, using a very limited repertoire of markers (SM22a). This study would be greatly improved by addition of single cell transcriptomics to determine the precise molecular phenotype of lineage-labeled cells and how this changes with postnatal lung maturation.

---

## [Author Response]

Reviewer #1:[…] 1) Since this is the first report of this new mouse model, additional characterization of its fidelity is warranted at each of the induction timepoints. The co-expression of the rtTA/GFP from this allele needs to be looked at more carefully including FACS analysis and additional immunohistochemistry to verify what% of total Pdgfra+ and other cell types it marks.

We have used flow cytometry analysis to better characterize our new mouse model. Data from flow revealed that the percentages of GFP+ cells in total lung cells are not statistically different in *Pdgfra^rtTA^;tetO-GFP* lungs as compared to *Pdgfra^GFP^* lungs (Figure 1D, E). Also, out of total PDGFRa+ cells as labeled by anti-CD140a (PDGFRa)-PE antibody in flow, 78.62% ± 5.71% were GFP+ in *Pdgfra^rtTA^;tetO-GFP* lungs, indicative of the efficiency of rtTA activation (Figure 1—figure supplement 1C, D).

2) The lineage tracing studies are interesting but not well defined. For instance, the labeling of Pdgfra+ cells at E9.5 could simply mark almost all lung mesenchyme/fibroblasts. If this is the case, then the interpretation of data in Figures 2 and 5 are somewhat unclear. The authors should provide an index at all of the times of dox induction (i.e. E9.5, P3, adult etc.) of what% of the total lung mesenchyme this allele marks. A comparison to a pan-mesoderm or mesenchyme marker would be helpful i.e. vimentin or something similar.

We added analysis of *Pdgfra+* lineaged cells at E12.5 following dox induction at E9.5 (Figure 2—figure supplement 1B, B’). We tried vimentin antibody as a pan-mesenchymal marker, but it did not give distinct mesenchymal staining in our hands. Nevertheless, it is easy to distinguish mesenchyme from epithelium at E12.5. We found that labeling at E9.5 does not label all mesenchymal cells (Figure 2—figure supplement 1B, B’). A similar density of labeling is observed in E12.5 *Pdgfra^GFP^* lungs, suggesting that incomplete labeling is a feature of *Pdgfra* expression in a subset of mesenchymal cells (Figure 2—figure supplement 1C, C’). The conclusion that *Pdgfra+* cells represents a subset of the lung mesenchymal cells is also supported by data from multiple other groups that have studied this cell population at various time points (Endale et al., 2017; Ntokou et al., 2015; Zepp et al., 2017).

3) In light of point 2 above, what do the authors make of the fact that SM22a and ADRP appear to mark the majority of alveolar mesenchyme at P7 (Figure 2)? In particular, ADRP seems to mark a lot of cells that are Pdgfra-. A better characterization of what these cells are is warranted.

Indeed, based on our data and prior published data from other labs, SM22a+ cells and ADRP+ cells are the two major cell types in the alveolar mesenchyme at P7. Our data showed that at P7, a majority of *Pdgfra*+ cells are myofibroblasts, rather than lipofibroblasts. For ADRP+ lipofibroblasts, previous reports showed that *Tbx4* lineage and *Fgf10* expressing cells contribute to lipofibroblasts (El Agha et al., 2014; Zhang et al., 2013). A more detailed reference to these data are added to the Introduction section.

4) The data in Figure 5 and Figure 5—figure supplement 1 are interesting in that they show a significant number of Pdgfra+ cells that proliferate in response to bleomycin injury. Are these the myofibroblasts or are they what appear to be a large number of SM22-/Pdgfra+ lineage traced cells?

Based on Zepp et al. (2017), it was shown that in the bleomycin model, SMA+ myofibroblasts do express Ki67, a proliferation marker. This suggests that proliferating cells can simultaneously have myofibroblast characteristics following injury. In our *Pdgfra*-lineaged lungs, as labeling was carried out prior to injury when the *Pdgfra*+ cells are SM22-, we deduce that the *Pdgra*+ proliferating cells originated from SM22- cells. We clarified this in Results in subsection “PDGFRa-lineaged cells contribute to pathological myofibroblasts in the bleomycin model of lung fibrosis” and in the Discussion section.

5) The IHC data shown in Figure 5 appear to show that SM22+ cells at day 14 are unlikely to be derived from Pdgfra-lineage. Is this due to early vs late contribution to bleomycin triggered myofibroblasts?

We added arrowheads in Figure 5G to highlight that at D14, a portion of the SM22+ cells are derived from the *Pdgfra* lineage. Quantification indicated that ~26% of SM22+ cells are from the *Pdgfra*+ (Figure 5O). This number increased to ~45% at D21.

6) The cell ablation experiment in Figure 3 suggests that the loss of Pdgfra-derived early postnatal myofibroblasts results in alveolar simplification. While the authors note that AT2 cell numbers are reduced, the percentage to total lung cells is unchanged. It is unclear how the latter conclusion was made or whether total cell number or epithelial cells specifically, were altered.

The percentage was calculated by SPC+ AT2 cells in total lung cells base on DAPI staining. Total cell numbers were decreased per area as the lung is simplified. These numbers were added to revised Figure 4 (formerly Figure 3) source files.

Reviewer #2:[…] Given the novelty of this mouse model and its potential importance to address relevant questions in the field, the details about the characterization of this line are relatively brief and sometimes missing. For example, they use two different regimens of Dox administration (IP and dox-containing food) but do not comment on the rationale and whether there were differences in labeling efficiency.

For prenatal lineage tracing, we chose dox containing food because it is gentle on the pregnant females and effective at inducing activity. For postnatal lineage tracing, we chose dox injection followed by dox food because it showed a much higher efficiency compared to dox containing food only, presumably because the pups do not get to the dox food themselves and the dox transfer through the milk from the dam is not efficient. This rationale is added to Materials and methods section.

What was the frequency of dox-independent labeling in these mice?

Very little leaky labeling without dox, and this result is added to Figure 2—figure supplement 1A.

Is labeling present in smooth muscle elsewhere in airways or large vessels?

We did not observe labeling in the airway or vascular smooth muscles. Labeling subjacent to the airway does not overlap with airway smooth muscle upon close inspection. This result was added to Figure 2—figure supplement 1J-L.

As a knockin line, Pdgfra^rtTA^;tetO-cre mice are haploinsufficient for Pdgfra. The authors should provide evidence or comment whether this may have influenced any of the results reported here, including the differences in labeling from that reported in the BAC transgene line.

This is an excellent point, and we added to the Discussion that this may be an additional possible distinction that may explain the difference between our data and the data from BAC transgenics. However, there is no notable phenotype in the *Pdgfra* heterozygotes to indicate haploinsufficiency. In addition, conditional *Pdgfra* loss-of-function mutants show decreased myofibroblasts and maintenance of lipofibroblasts (McGowan and McCoy, 2014). Thus, if there is any undetected haploinsufficiency, we would expect that using our *Pdgfra^rtTA^* knockin line, there should be fewer myofibroblasts labeled compared to the BAC transgenic line, opposite to what the data show.

The surprising and apparently paradoxical high number of myofibroblasts Pdgfra-lineage labeled negative in the BPD-hyperoxia model is one of the most interesting observations in the paper. The conclusion that this may have been accounted for by lipofibroblasts transdifferentiating into myofibroblasts should be explored in this model by lineage-labeling lipofibroblasts (ADRP or other relevant cre line). This will strengthen their conclusion and will offer further support to their Pdgfra^rtTA^;tetOcre mouse model.

PLIN2 (ADRP) is the most commonly used lipofibroblast marker in lung. There is a published cre line, *Plin2 (Adrp*)^creERT2^ (El Agha et al., 2016; Ntokou et al., 2017). However, it is not specific for lipofibroblasts, and labels a number of other cell types in the lung (El Agha et al., 2016). Thus, we currently do not have the tool to specifically trace the lipofibroblast lineage. We added in Discussion that a high priority next-step is to identify markers that are specific for lipofibroblasts, and use them to generate mouse tools for lineage tracing and address the potential of lipofibroblasts in development and injury.

Reviewer #3:This manuscript by Li et al. describes the generation of a new line of mice in which rtTA is placed under the regulatory control of the endogenous Pdgfra promoter. This new line was validated by comparing specificity of Dox-dependent tetO-GFP expression with expression of a GFP reporter in lungs of Pdgfra^GFP^ mice. Authors go on to show that Pdgfra-lineage cells of the embryonic lung transiently generate SM22a-immunoreactive cells in the perinatal period which then contribute to lipofibroblasts, that ablation of Pdgfra-expressing cells in the perinatal period leads to alveolar simplification and that Pdgfra-lineage cells differentially contribute to SM22a-immunoreactive cells in models of neonatal hyperoxia and adult bleomycin-induced fibrosis. Strengths of this study are the demonstration of a valuable new genetic tool to investigate the biology of Pdgfra-expressing cells of the lung and application of this tool to study the dynamics of lung mesoderm during development and perinatal lung maturation. A concern with this study is the broad classification of SM22a+ cells of the developing, perinatal and adult lung as myofibroblasts. Lineage tracing of Pdgfra-expressing cells is performed at various times of embryonic or postnatal lung maturation which complicates interpretation of data since Pdgfra is dynamically expressed within mesenchymal cells during these stages of lung development/maturation.

We added analysis with a-SMA staining which substantiated the findings with SM22a (Figure 2—figure supplement 1D-F). We added in the Introduction a more specific definition of myofibobroblasts in the normal lung. Findings from our prior study (Branchfield et al., 2016) indicated that a-SMA and SM22a expression is absent in the distal mesenchyme in the prenatal lung. Their expression increases and peaks during the first phase of alveologenesis (P3-14), and then subsides to a low level afterwards and into adulthood. Thus, the term myofibroblasts in the normal lung only refers to the a-SMA+ and SM22a+ cells during alveologenesis when secondary septae/crests are added. Some have also referred to these cells as secondary crest myofibroblasts which is a more distinct term.

We performed lineage tracing at both prenatal (E9.5) and postnatal (P0) stages, and analyzed data at E12.5, P7 and P40 to best depict the dynamic nature of the *Pdgfra*+ cell population.

Specific concerns include:1) The conclusion that ".hitherto unappreciated complexity in the behavior of the Pdgfra-lineaged cells" seems overstated. Previous studies by this group and others demonstrate that the molecular phenotype of lung mesoderm is dynamically regulated during development.

We have deleted “hitherto unappreciated”.

2) Results subsection “Lineage tracing revealed that Pdgfra cells give rise primarily to normal alveolar myofibroblasts”: “[…] demonstrate that in the early postnatal lung, Pdgfra-lineage cells give rise primarily to myofibroblasts, with minor contributions to lipofibroblasts and endothelial cells.” This statement is a little misleading in that it implies that myofibroblast and lipofibroblast populations are distinct and fixed. In reality, as shown previously by this group (Branchfield et al., 2016) and nicely demonstrated in this study, these populations are dynamic and are either interchangeable or their relative populations change through "pruning" during early postnatal lung maturation. The authors provide evidence supporting the notion that these populations are interchangeable based upon long-term retention of lineage-labeled cells. A concern with these experiments is that all conclusions are made using immunofluorescence analysis to evaluate fibroblast phenotype in histologic sections, using a very limited repertoire of markers (SM22a). This study would be greatly improved by addition of single cell transcriptomics to determine the precise molecular phenotype of lineage-labeled cells and how this changes with postnatal lung maturation.

We thank the reviewer for these suggestions and have added single-cell transcriptomic data of *Pdgfra*-GFP+ cells at both P7 and P15 (new Figure 3). The scRNAseq data support our lineage tracing findings and provide further resolution of the complexity within the *Pdgfra*+ cells (see Results subsection “Single cell transcriptome analysis of *Pdgfra+* cells revealed diversity during

alveologenesis” and the Discussion section). We have also added analysis with a-SMA staining which substantiated the findings with SM22a (Figure 2—figure supplement 1D-F). By using *Pdgfra*-based lineage tracing in the normal lung starting at both prenatal and postnatal time points, we have observed labeling primarily in secondary crest myofibroblasts. The lineaged cells are not lost past alveologenesis due to pruning as suggested. They stay in the lung mesenchyme and simply turn down their SMA and SM22 expression. They do not interchange into lipofibroblast cells.